# COVID-19 Vaccine Hesitancy among U.S. Veterans Experiencing Homelessness in Transitional Housing

**DOI:** 10.3390/ijerph192315863

**Published:** 2022-11-29

**Authors:** June L. Gin, Michelle D. Balut, Aram Dobalian

**Affiliations:** 1Veterans Emergency Management Evaluation Center, U.S. Department of Veterans Affairs, 16111 Plummer St. MS-152, North Hills, CA 91343, USA; 2Division of Health Services Management and Policy, The Ohio State University College of Public Health, 250 Cunz Hall, 1841 Neil Ave, Columbus, OH 43210, USA

**Keywords:** homeless persons, COVID-19 vaccine, vaccine hesitancy, veterans, health behaviors

## Abstract

Little is known about COVID-19 vaccine hesitancy and acceptance among individuals experiencing homelessness, despite their higher risk for morbidity and mortality from SARS-CoV-2. This study examines COVID-19 vaccination attitudes and uptake among U.S. military Veterans experiencing homelessness enrolled in transitional housing programs funded by the U.S. Department of Veterans Affairs (VA). Telephone interviews were conducted with 20 Veterans in California, Florida, Iowa, Kentucky, and Massachusetts, USA (January–April 2021). A rapid analysis approach was used to identify and enumerate commonly occurring themes. Although 60% of interviewed Veterans either received the COVID-19 vaccine or were willing to do so, one-third expressed hesitancy to get vaccinated. COVID-19 vaccination attitudes (e.g., belief that the vaccines were inadequately tested), military experience, beliefs about influenza and other vaccines, and sources of information emerged as influential factors for COVID-19 vaccination uptake or hesitancy. Veterans in VA-funded homeless transitional housing programs are generally willing to be vaccinated. However, a substantial minority is reluctant to take the vaccine due to concerns about the COVID-19 vaccine and distrust of authority. Recommendations for increasing uptake include utilizing Veteran peers, homeless service providers, and healthcare providers as trusted messengers to improve confidence in the vaccine.

## 1. Introduction

The COVID-19 pandemic has brought disruption and the risk of illness and death for individuals living in institutional settings throughout the United States. While vaccines offer hope for a return to safety and normalcy, COVID-19 vaccine hesitancy among socioeconomically disadvantaged individuals is poorly understood. Recent research has begun closely examining vaccine acceptance among vulnerable populations [1,2,3]. Globally, people experiencing homelessness and migrants are particularly vulnerable to vaccine mistrust while also facing especially high risks for contracting COVID-19 because of their reliance on congregate settings for shelter or other social services, high rates of pre-existing health conditions, and the highly transient nature of homeless living [4,5,6,7].

Promoting COVID-19 vaccine uptake among people experiencing homelessness is essential to reducing transmission, illness, and death. For these reasons, the U.S. Centers for Disease Control and Prevention (CDC) issued guidelines advising that state and local health departments consider prioritizing the vaccination of homeless populations residing in congregate facilities [8]. However, COVID-19 vaccine uptake among homeless populations may be hindered by myriad factors, including low motivation, lack of early prioritization, vaccine hesitancy, and barriers such as technology and transportation that preclude them from accessing vaccines [9,10]. Additionally, the spread of misinformation and antivaccination movements have led to vaccine hesitancy among the general U.S. adult population [11]. People experiencing homelessness may be especially susceptible to misinformation due to experiences of stigma and racism in healthcare settings [12,13].

Previous studies examining COVID-19 vaccine hesitancy and uptake among individuals and Veterans experiencing homelessness in the U.S. [14,15,16,17,18] and abroad [6,7] reported that around 45% had been fully vaccinated for COVID-19, with COVID-19 vaccination levels in the homeless community lagging an estimated 15–25% behind the general population [16,19,20]. Limited research on general vaccine hesitancy [21,22,23] suggests that vaccine refusal among homeless populations tends to be high even for longstanding illnesses such as hepatitis and influenza. These studies call for more research to better understand COVID-19 vaccine hesitancy and uptake behavior among people experiencing homelessness to identify root causes and ways of improving vaccine acceptance.

One subgroup of people experiencing homelessness, homeless Veterans, may differ from the general homeless population in vaccine attitudes and behaviors due to their unique experience of serving in the military. Prior to the U.S. military vaccine mandate [24,25], active-duty military members were refusing the COVID-19 vaccine in higher proportions than the general population [26], suggesting that military experience may be relevant to shaping vaccine uptake behavior [27]. Mistrust toward government health recommendations is prevalent among unhoused persons in general [12,19,20,28], and unhoused Veterans specifically [27,29], possibly influencing their vaccine attitudes and uptake.

Ensuring that Veterans experiencing homelessness receive the vaccine in significant numbers is important to the U.S. Department of Veterans Affairs’ (VA) effort to reduce the spread of COVID-19, as Veterans are disproportionately represented within the U.S. adult homeless population [30,31]. This study sought to understand COVID-19 vaccination attitudes and behaviors of Veterans enrolled in VA Grant and Per Diem (GPD) programs. Through the GPD program, VA funds non-VA entities to provide transitional housing and services to Veterans experiencing homelessness. Many of these organizations also provide housing and social services to the general non-Veteran homeless population. Veterans in the GPD program may receive up to two years of transitional housing, during which they are expected to undertake steps to achieve economic self-sufficiency and find permanent housing [30,32,33,34]. This study aims to understand their attitudes, willingness, barriers, and facilitators toward COVID-19 vaccination, which may yield insights into ways to reduce vaccine hesitancy and barriers and improve vaccine acceptance within this population.

## 2. Materials and Methods

### 2.1. Recruitment & Data Collection

Qualitative interviews were conducted with Veterans from five GPD organizations in California, Florida, Iowa, Kentucky, and Massachusetts. The GPD organizations were all male-only facilities. The research team invited GPD grantee organizations to participate in the study through a monthly webinar call hosted by the National VA GPD Office, which is attended by GPD grantee organizations and VA homeless program staff. GPD grantee organizations interested in participating were invited to contact the research team. Seven organizations contacted the research team to participate in the study. These GPD organizations were then encouraged to invite their Veteran residents to participate in one of two ways: (1) Veterans received a recruitment flyer, which included a phone number they could use to contact the research team to set up a time for an interview, or (2) GPD organizations collected the names and phone numbers of Veteran residents who expressed interest in participating, and with the Veterans’ consent, shared that information with the research team, who contacted the Veterans directly. Five of the seven organizations successfully recruited Veterans to participate in interviews. The research team also interviewed staff at all seven GPD organizations to inquire about their experiences providing care during COVID-19 and what they were doing to facilitate Veterans’ vaccination. Staff perspectives are reported elsewhere [35,36].

The study team conducted qualitative, semi-structured telephone interviews lasting approximately 30 min from January to April 2021 until data saturation was reached [37], resulting in interviews with 20 Veterans. Verbal consent to participate was obtained prior to study inclusion. The only inclusion criterion was that the individual be a Veteran currently enrolled in a GPD organization. Veterans did not receive compensation for participation in the study. This study was reviewed by the VA Greater Los Angeles Institutional Review Board and determined to be a quality improvement study (Project Number: 2021-000413).

The interview guide was developed by the research team and consisted of 20 open-ended questions. The interview guide was derived from prior studies of vaccine attitudes and behavior among people experiencing homelessness [22,23], whose questions were, in turn, derived from the World Health Organization and the SAGE Working Group on Vaccine Hesitancy [38]. The Interview Questionnaire Tool can be found in Appendix A. During the interviews, Veterans were asked to describe their views about the COVID-19 vaccine and their reasons for either hesitation or interest in being vaccinated. They were also asked about their concerns about the novel coronavirus, their sources of information about the COVID-19 vaccine, their trust in healthcare providers, their vaccination history, and their sociodemographic characteristics. Participants were asked a six-point Likert-scale question, which inquired about the respondents’ likelihood of getting the COVID-19 vaccine when it became available. Veterans could respond “definitely not”, “probably not”, “don’t know/neutral/don’t have enough information”, “probably yes”, “definitely yes”, or “already vaccinated”.

### 2.2. Data Analysis

All interviews were audio recorded and transcribed. The study team utilized a rapid analysis approach (April to May 2021) to analyze the interview transcripts [39,40] using inductive grounded theory [41], identifying themes that were prominently expressed in Veterans’ perspectives about vaccination and within the deductive domains that were identified a priori [41,42]. First, a templated summary table of key domains was developed based on the interview guide. The summary table was reviewed and modified after being tested by the analytic team to summarize a single transcript, which was used to identify broad thematic categories to guide analysis. Using the templated summary table, the 20 transcripts were divided, and two team members independently summarized them, grouping information by shared content. Two summaries were then randomly selected to undergo a secondary review by the other team member to ensure consistency in the data being recorded. Summaries were then consolidated into a single document to identify commonly occurring themes across all interviews. The substantive significance of themes was used to ascertain their significance [43]. Substantive significance refers to the extent and context in which these themes were present in the data. We measured the presence of themes in the data both qualitatively, by examining their context, and quantitatively, by recording the number of individuals that mentioned each theme.

### 2.3. GPD Sites

Participating organizations varied in size and housing format, including two site-based programs with GPD beds located within larger organizations also serving the general homeless population, and three scattered-site programs with Veterans living in housing separate from the larger organization. Given the variation in timing, some Veterans were interviewed before the COVID-19 vaccine was first offered to them, whereas others were interviewed after they had access to vaccines.

## 3. Results

Respondents were between 29 to 65 years old, with most being in their 40s to late 50s. The majority identified were white, though five were people of color and four declined to state their race. Some GPDs made vaccines available onsite to Veterans through the local health department and the local VA, whereas Veterans at other sites had to go to the VA to receive the vaccine. Veterans mentioned factors that shape their vaccination decisions, including the rapid development of the vaccine and their views overall toward the COVID-19 pandemic. Major themes that emerged from the interviews included: (1) COVID-19 vaccination uptake attitudes; (2) Veterans’ military experiences; (3) influenza vaccination uptake; and (4) sources of information about the COVID-19 vaccine.

### 3.1. COVID-19 Vaccination Uptake Attitudes, Behaviors, Barriers, and Facilitators

Figure 1 presents the likelihood of COVID-19 vaccination among the study participants. Eight Veterans (40%) had been vaccinated by the time of their interview, while four others (20%) were eager or willing to be vaccinated but were still waiting for the opportunity, and two (10%) were leaning toward acceptance. However, six Veterans (30%) expressed hesitancy or unwillingness to take it. Combining those already vaccinated with those who would “definitely” get it, 60% of the Veterans would likely uptake the vaccine when widely available without a need for further intervention to persuade or facilitate it.

Veterans stated various reasons for being either willing or hesitant to be vaccinated. Figure 2 shows the breakdown of how many Veterans mentioned each reason as influential in their decision whether to uptake or refuse the vaccine (note: Veterans may have cited more than one reason).

Veterans who were eager or willing to uptake the vaccine had rationales that converged around the core theme of wanting to prevent illness, death, and transmission to loved ones and their community:
“My first wife died from [COVID-19] and thousands of Americans are dying from it. I got the shot, I got PTSD, and it gave me some nightmares… I had flashbacks of being in the jungle again… I’m still going to get the second shot though, because… through time, I’ll be pretty much protected from it.” (Iowa2)

Veterans who were more hesitant to get the vaccine had slightly more varied rationales. Perceptions that vaccines were either insufficiently tested or were “too new” were mentioned most frequently. Among vaccine-reluctant Veterans, all but one noted that the rapidness with which the COVID-19 vaccines were approved left too much uncertainty for their comfort. Half of the vaccine-reluctant Veterans also mentioned the related concern that long-term side effects of the COVID-19 vaccines were still unknown:

“They rushed this in like seven, eight months, so they can’t tell you what it’s going to be like in two years, five years. You know, back when the smallpox [vaccine] first came out, they said—‘Oh, that was fine.’ Then they started having birth defects and you know, like different problems with that, you know what I mean?… You can’t tell me what it’s going to be like five years down the road, just to be safe today. I don’t want to get cancer five years down the road, so I didn’t get COVID today.” (Florida2)

Mistrust of either government or vaccine manufacturers was also a factor in more than half of the vaccine-reluctant Veterans’ decision. The youngest Veteran was hesitant to get vaccinated because he does not like needles, while another did not see the need to get the vaccine since he already had COVID-19.

All 20 Veterans unanimously reported that they did not experience or anticipate any barriers to accessing the COVID-19 vaccine. Two of the five organizations facilitated on-site vaccination. Two Veterans noted that while they were willing to be vaccinated, their motivation was low given the competing demands, such as finding permanent housing, that for a homeless person supersede the urgency of vaccination:

“If someone wants to give it to me, I would absolutely, 100% take it. No issues, whatsoever. But… it’s such a low priority, I’m not generally going to seek it out.” (Iowa1)

Three others noted that they would gladly get the vaccine if it were required for jobs or other benefits. Even among those who are very open to vaccination, Veterans experiencing homelessness may fail to uptake the vaccine unless it is available to them with little to no effort.

### 3.2. Military Experience

Veterans’ military experiences and identity emerged as influential in shaping their views toward vaccines and had both a positive and negative effect on their vaccination uptake attitudes. Some Veterans mentioned their distrust of government or the military as their reason for refusing to get the vaccine; however, most Veterans who cited getting mandatory vaccines regularly as part of their military experience noted that they felt acclimated to getting recommended vaccines as a result:

“They said you can get a vaccine today, and I said okay… When somebody offers me a vaccine, I’m like all right I’ll get it. I’m prepared… I’m used to being a guinea pig… I was in the Army active for three and a half years, and then I did nine and a half years total, the reserve and active. So, you know people sticking needles in me, telling me, ‘all right you need to take this’, and I’m like ‘okay’.” (Massachusetts4)

The significance of military experience was especially notable given that despite not being asked about it, 11 of the 20 Veterans mentioned their military experience as influencing their views toward the vaccine. A quarter of Veterans noted that many immunizations were required in the military, such as the influenza or anthrax vaccines. Two recently discharged Veterans noted that there were adverse career consequences to not getting these vaccines during military service.

For some Veterans, military experience engendered a sense of distrust toward government, the VA, and the military that extended to vaccine refusal. Exposure to environmental hazards during military service was cited as another factor behind distrust of government-endorsed science. One Veteran cited his exposure to water contamination at the Camp Lejeune Marine Base as a reason for distrust of the COVID-19 vaccine. Another Veteran cited mandatory vaccination while serving in the military and wanted to avoid putting more unnecessary things into his body now that he had a choice:

“Going into the military, I had tons of vaccines, and putting one more in my body is something I try not to do if I don’t have to.” (California2)

However, not all Veterans who expressed mistrust of the military and government were hesitant to receive the COVID-19 vaccine. One emphasized their desire to avoid spreading COVID-19 to loved ones helping to overcome feelings of military and government distrust.

### 3.3. Influenza Vaccine Uptake

Most Veterans who refused to get the COVID-19 vaccine said their concerns about this specific vaccine made them hesitant. Only one expressed skepticism and hesitancy about vaccines in general; the remaining Veterans, including five of those who did not intend to get the COVID-19 vaccine, affirmed their beliefs that vaccines are effective tools to prevent disease. When asked whether they recently got the influenza vaccine, nine of the 20 Veterans received the flu shot in either 2019 or 2020. All but one of the Veterans who received the flu vaccine had either gotten the COVID-19 vaccine or intended to get it, suggesting among flu shot recipients a tendency to be more accepting of vaccines. A few Veterans had also received vaccines for hepatitis, and one reported having to delay COVID-19 vaccination because of recent hepatitis vaccination.

Of the seven Veterans who reported that they routinely refuse flu shots, half accepted the COVID-19 vaccine, suggesting higher demand for the COVID-19 vaccine than the flu vaccine given the strong virulence and seriousness of COVID-19. Some Veterans cited distrust in both flu and COVID-19 vaccines.

### 3.4. Sources of Information

Veterans reported getting their information about COVID-19 and the vaccine from a wide variety of sources, with some citing more than one primary information source, see Figure 3.

The “news” was the most popular, comprising the main source of information for 14 of the Veterans. Of these 14, nine were willing to get vaccinated, while four were unwilling, and one was undecided but leaning toward yes. “News” included sources ranging from television news and physical newspapers to podcasts, with television news comprising the vast majority (8 of 14). Eight Veterans, half of whom also had “news” as an information source, mentioned the internet as their primary way of seeking information. Internet sources included videos on YouTube and social media (Facebook or Instagram), but also, for younger Veterans, Reddit and stock market apps.

Seven Veterans said they primarily receive their information on the COVID-19 vaccines from peers, including family, friends, colleagues, or strangers. Some Veterans said they trusted the word of close others in their lives more than “some news anchor’s opinion” (Iowa1) or social media:

“The most reliable source of information about COVID is direct word of mouth from someone who has had it.” (Kentucky2)

Of these seven, three were already vaccinated, three were unwilling, and one was undecided but leaning toward yes. Nine Veterans had spoken to other GPD Veterans about the vaccine: some were hesitant, while others were willing to be vaccinated. Two Veterans reported being convinced to take the vaccine or having convinced others to get it through these conversations.

Eight Veterans cited information from professionals, including GPD staff, healthcare workers, or government organizations, as their primary information sources. Of these, five were already vaccinated, two were unwilling, and one was undecided and leaning toward yes. The Veterans did not receive any educational information about the COVID-19 vaccine from their own physician, with 17 of the 20 Veterans explicitly reporting that no such vaccine information was given, even though 11 Veterans said they trust their provider. Two Veterans in their 30 s said they did not trust their VA doctors because of bad experiences. The GPD was cited as a trustworthy source by seven Veterans who received information about the vaccine from their GPD staff. Several Veterans noted that they were not sure what sources to trust, given the prevalence of misinformation on television and the internet.

## 4. Discussion

Vaccine reluctance has previously been linked to confidence (distrust), complacency (lack of perception of seriousness), and convenience (perceived barriers to access) [38,44,45], all of which are highly prevalent barriers among people experiencing homelessness. Most Veterans interviewed in this study were either willing or eager to be vaccinated, citing concerns about mitigating the worst sequelae of the pandemic. Many were especially concerned about avoiding spreading COVID-19 to their community, perhaps related to their dedication to protecting and serving their country. Prior research has found civic responsibility and a desire for normalcy as compelling reasons for COVID-19 vaccination among people experiencing homelessness [12]. Most reported that they strongly believed in other vaccinations, with previous influenza immunization a possible predictor of COVID-19 vaccination uptake. Similar findings were cited in previous studies examining the association between seasonal influenza and influenza A (H1N1) vaccine acceptance [46,47].

No access barriers were reported among Veterans. Two of the five GPD programs provided vaccines onsite in February 2021 when vaccine access was very limited. Some Veterans had assistance setting up appointments from VA and GPD staff, which helped facilitate uptake. Interventions to facilitate access have been previously recommended to reduce barriers and facilitate vaccination for people experiencing homelessness [12,13,16,20,48,49,50]. Convenient access was key to vaccine uptake for many Veterans, as several GPD residents noted that their lack of motivation or a sense of complacency. Such challenges to motivation have been estimated to hinder more people in the U.S. than vaccine skepticism [51]. A third of the interviewed Veterans were unwilling to be vaccinated. Distrust was the most prevalent reason for hesitation or refusal, with concerns expressed about the COVID-19 vaccine’s uniquely “new” nature, its rapid development, and unknown side effects. The unexpected significance of military experience in shaping homeless Veterans’ vaccine uptake perspectives warrants further investigation. The mandatory nature of vaccines in the military may prompt some Veterans to wait for stronger encouragement than strictly voluntary vaccine uptake. However, a COVID-19 vaccination mandate may compound the already high levels of distrust toward government and vaccine manufacturers, which has been commonly cited among unhoused individuals refusing the COVID-19 vaccine [12,19,20,28]. However, only a few said that their mistrust led to refusal of the vaccine, suggesting that vaccine-hesitant Veterans uniquely connect their general feelings of mistrust to concerns about this vaccine. Accordingly, peer education would likely be more effective for Veterans experiencing homelessness than a mandate and may help disentangle vaccine-related distrust from distrust stemming from factors such as environmental contamination or mistreatment. Further, incentives and societal benefits may encourage vaccination uptake decisions.

Some support was found for Kuhn’s [52] finding that homeless individuals who relied on unofficial sources such as word of mouth tended to be more vaccine hesitant than those relying on official sources. GPD Veterans who primarily relied on news media for information were mostly willing to accept the vaccine. However, most Veterans reported relying on multiple sources—both official and unofficial. Several vaccine hesitant Veterans expressed skepticism toward the news media, while many Veterans noted that they most trusted information from their healthcare providers, fellow Veterans and GPD staff. Notably and despite VA outreach efforts including automated calls, Veterans reported receiving little to no educational information during this early vaccine rollout period from their healthcare providers. However, clinicians providing care to Veterans in VA’s Homeless Patient Aligned Care Team (HPACT) clinics have reported providing information about the COVID-19 vaccines to their patients [35]. Veterans’ physicians, GPD staff members, behavioral health providers, and other health professionals might be a promising avenue for encouraging COVID-19 vaccination. These professionals are likely to be trusted messengers to other homeless populations as well [13,49,53,54]. Particularly, the role of transitional housing providers in promoting vaccine uptake among people experiencing homelessness warrants greater attention. Peers, including other unhoused persons [19,53], or other Veterans [27,29], can be an especially important trusted source that Veterans in VA homeless programs rely on for COVID-19 vaccine decisions.

### Limitations

GPD program-enrolled Veterans differ from the general homeless population in that most have access to a regular source of health care and a safe and secure living space [55], better health-seeking behaviors and reducing the daily worry of where they will sleep and find food each day [56]. Thus, these advantages enabled GPD-enrolled Veterans to be more likely to be vaccinated than others experiencing homelessness, as Balut [14] found. These findings are likely more generalizable to populations of transitional housing residents than others experiencing homelessness. Additionally, GPD-enrolled Veterans are almost exclusively men, and exclusively those who have served in the U.S. military. Military experiences and identities may shape their attitudes toward vaccines in ways that may not be generalizable to other populations experiencing homelessness. Distrust of authority is also prevalent among the general homeless population, due to experiences of stigma and racism while receiving health care [57,58,59]. In addition, the convenience sampling recruitment method may have led to selection bias. Two of the seven participating organizations did not successfully recruit Veterans, possibly reflecting additional vaccination recruitment barriers that the study could not capture. This study did not inquire about Veterans’ political affiliation, which has been shown to heavily influence COVID-19 protective measures in the general adult population, including vaccination [60]. The sample size precluded examining associations between age, race/ethnicity, and COVID-19 vaccination status. Lastly, respondents were interviewed when the Pfizer-BioNTech vaccine was only being administered under an Emergency Use Authorization (prior to its full authorization) by the U.S. Food and Drug Administration, and before the vaccines were widely available to the public, so we were not able to assess whether these events influenced uptake over time.

## 5. Conclusions

Low vaccination rates among people experiencing homelessness pose a challenge for congregate residential programs due to concerns about COVID-19 transmission. Increasing homeless Veterans’ uptake of the COVID-19 vaccine would ameliorate safety concerns related to shared housing in VA-funded GPD facilities. VA Homeless Program documents [61] suggest that concerns about infection and spread of COVID-19 were significant among GPD organizations housing Veterans experiencing homelessness, with some organizations suggesting that they were weighing whether to make vaccination mandatory for residents [35]. Finally, increasing vaccination uptake in homeless individuals is critical to racial and social equity, as the burden of both homelessness and COVID-19 infections fall disproportionately on African Americans, Latinos, and other people of color [62,63].

People experiencing homelessness are, like most of the population, highly aware of the risks of COVID-19, and are likely willing, if not eager, to be vaccinated when opportunities are easily available. However, these findings underscore the vital role of trusted others as conduits for communicating the risks and benefits of vaccination to homeless individuals with vaccination doubts. Peer education and outreach by healthcare and homeless service providers, accompanied by easy access to vaccines, are likely to be highly influential in increasing vaccine acceptance and uptake. Understanding hesitancy toward vaccination uptake in vulnerable populations, such as Veterans experiencing homelessness, is vital to ultimately reducing the spread of COVID-19 and mitigating the pandemic’s worst impacts on unhoused individuals.

## Figures and Tables

**Figure 1 ijerph-19-15863-f001:**
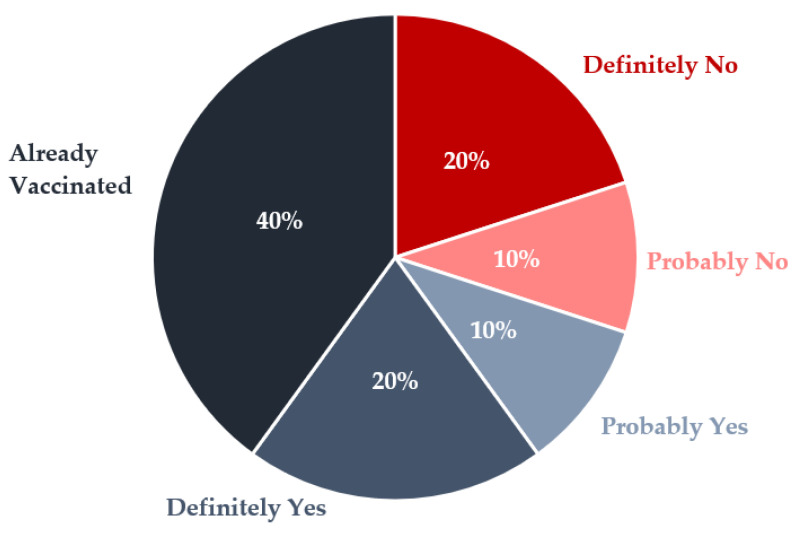
Likelihood of COVID-19 Vaccination among GPD Veterans.

**Figure 2 ijerph-19-15863-f002:**
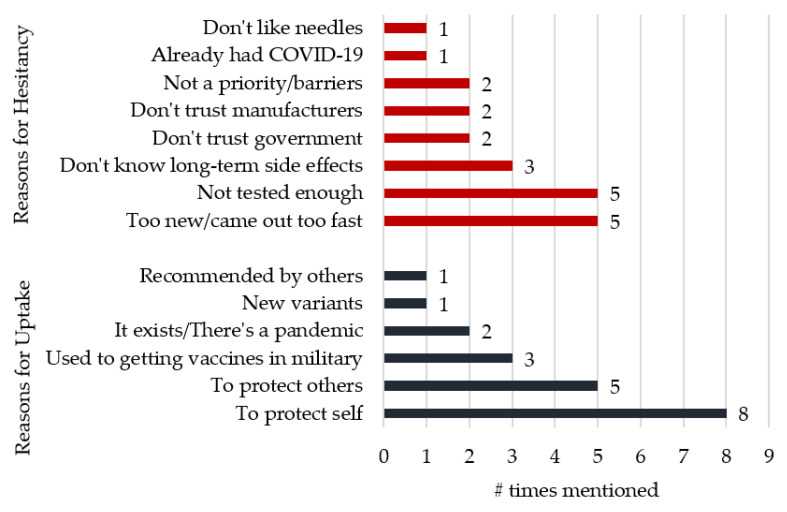
Reasons Given for COVID-19 Vaccine Willingness or Hesitancy.

**Figure 3 ijerph-19-15863-f003:**
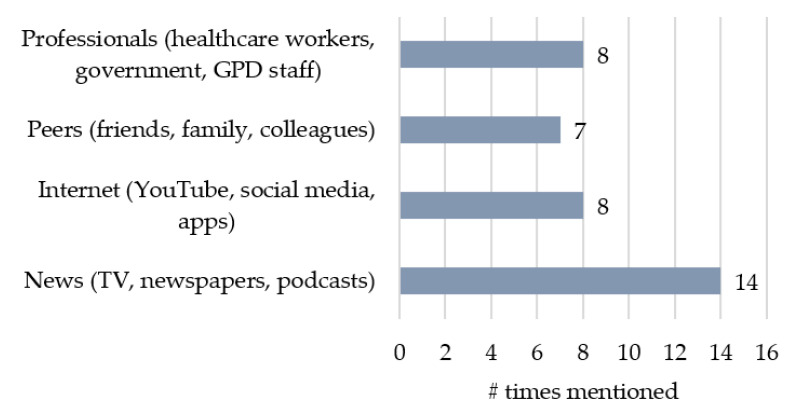
Sources of Information on COVID-19 and COVID-19 Vaccines.

## Data Availability

The datasets generated and/or analyzed during the current study are not publicly available. They are available from the corresponding author on reasonable request, subject to approval from the ethics committee that approved the study.

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
