# Peer review of "COVID-19 Vaccine Hesitancy among U.S. Veterans Experiencing Homelessness in Transitional Housing"

_ijerph, 2022, doi:10.3390/ijerph192315863_

Round 1

Reviewer 1 Report

Thank you very much for the opportunity to review this article by Gin et al. regarding the COVID-19 vaccine hesitancy among a vulnerable group of U.S. Veterans.

I have just two minor comments:

1-     The author did not clarify the validity of the questionnaire. Has it been validate before or did they do any validation exercise? Also, it will be good to add the 20 questions as supplementary table, so other researcher can use it.

2-     One strength of this study is that it was done among vulnerable group. However, the author did not discuss their results with other studies among other vulnerable groups. Only mentioned it in the conclusion. It would add great value to the manuscript if the author can add a small paragraph and refer to other recently published studies in this regards. This is an example (DOI: 10.3390/vaccines10101634).

3-     The similarity report shows high similarity, 7% with one article (https://doi.org/10.1080/21635781.2022.2123415). I think this needs to be corrected.

Author Response

1-The author did not clarify the validity of the questionnaire. Has it been validate before or did they do any validation exercise? Also, it will be good to add the 20 questions as supplementary table, so other researcher can use it. Response: Thank you for highlighting the need to speak to the validity of the interview instrument. We added the following sentence to the Materials and Methods section on page 3, lines 111-114, to explain the basis of validation for the questions in our interview guide: The interview guide was derived from prior studies of vaccine attitudes and behavior among people experiencing homelessness [22,23], whose questions were, in turn, derived from the World Health Organization and the SAGE Working Group on Vaccine Hesitancy [38]. Because these prior studies of vaccine attitudes and behavior have provided support for the face validity of these research questions, we based our interview guide on these same questions. Following the suggestion, we have also added a supplemental Word file “Supplement 1” containing these questions. 2- One strength of this study is that it was done among vulnerable group. However, the author did not discuss their results with other studies among other vulnerable groups. Only mentioned it in the conclusion. It would add great value to the manuscript if the author can add a small paragraph and refer to other recently published studies in this regards. This is an example (DOI: 10.3390/vaccines10101634). Response: Thank you for recognizing the strength of this research examining vulnerable groups. We have added the following language to the Introduction, page 1, highlighting recently published studies examining vaccine acceptance among vulnerable populations, particularly in a global context involving people experiencing homelessness and migrant populations: Recent research has begun closely examining vaccine acceptance among vulnerable populations [1-3]. Globally, people experiencing homelessness and migrants are particularly vulnerable to vaccine mistrust while also facing especially high risks for contracting COVID-19 because of their reliance on congregate settings for shelter or other social services, high rates of pre-existing health conditions, and the highly transient nature of homeless living [4-71,2]. 3- The similarity report shows high similarity, 7% with one article (https://doi.org/10.1080/21635781.2022.2123415). I think this needs to be corrected. Response: We regret our oversight regarding the overlapping text between the submitted paper and our other published work: June L. Gin, Michelle D. Balut & Aram Dobalian (2022) Vaccines, Military Culture, and Cynicism: Exploring COVID-19 Vaccination Attitudes among Veterans in Homeless Transitional Housing, Military Behavioral Health, 10:4, 451-459. We would emphasize that while both papers are using the same study data, each paper has a different focus and address different research questions. The introduction, results, and discussion completely differ between the two papers. Most of the matching text occurred in the Data Analysis subsection within the Materials and Methods section, since both papers are products of the same research study. To remedy repetition, we have re-worded three sentences in the Data Analysis subsection. There was also some text matching in the first two sentences of the Results section describing sample demographics in both this paper and the published article, as both papers are based on the same data from the sample of Veterans. To remedy this, we reworded the first two sentences of the Results section to avoid redundancy.

Reviewer 2 Report

The research is interesting and the quality of presentation is high. The only fault I may notice to this research is the small number of participants - 20. Still the research has its value as a quality improvement study and I recommend  the publication in this special issue.

Author Response

Reviewer 2: The research is interesting and the quality of presentation is high. The only fault I may notice to this research is the small number of participants - 20. Still the research has its value as a quality improvement study and I recommend the publication in this special issue. Response: Thank you for your recognition of our study’s value as a quality improvement study. As a qualitative study, we found that we reached data saturation after interviews with 20 Veteran participants (Materials and Methods section, page 3, lines 102-104). Although a seemingly small sample, data saturation (capturing the majority of themes in a homogenous sample) can be typically achieved at 6-7 interviews (see https://journals.plos.org/plosone/article?id=10.1371/journal.pone.0232076 ), and 12 interviews can achieve higher degrees of saturation. Existing literature indicates that little new information is gained as sample sizes approach 20 interviews. However, due to the variability in our population of vulnerable Veterans, we opted to interview 20 participants to fully capture the nuances of their varied experiences, indicating an extremely high degree of data saturation.

Reviewer 3 Report

A

Despite is an interesting work the small sample size is insufficient. Because the primary goal of inferential statistics is to generalize from a sample to a population, authors must consider to include more participants in their study.

Author Response

Reviewer 3: Despite is an interesting work the small sample size is insufficient. Because the primary goal of inferential statistics is to generalize from a sample to a population, authors must consider to include more participants in their study. Response: Thank you for your interest in our work. As a qualitative study, we found that we reached data saturation after interviews with 20 Veteran participants (Materials and Methods section, page 3, lines 102-104). Although a seemingly small sample, data saturation (capturing the majority of themes in a homogenous sample) can be typically achieved at 6-7 interviews (see https://journals.plos.org/plosone/article?id=10.1371/journal.pone.0232076 ), and 12 interviews can achieve higher degrees of saturation. Existing literature indicates that little new information is gained as sample sizes approach 20 interviews. However, due to the variability in our population of vulnerable Veterans, we opted to interview 20 participants to fully capture the nuances of their varied experiences, indicating an extremely high degree of data saturation.
